# ADAPTIVE AND GENERATIVE ZERO-SHOT LEARNING

**Yu-Ying Chou**[1,3]**, Hsuan-Tien Lin**[3] **& Tyng-Luh Liu**[1,2]

[1]Institute of Information Science, Academia Sinica, Taiwan   [2]Taiwan AI Labs, Taiwan
[3]Department of Computer Science, National Taiwan University, Taiwan
{d07922014,htlin}@csie.ntu.edu.tw,liutyng@iis.sinica.edu.tw

## ABSTRACT

We address the problem of generalized zero-shot learning (GZSL) where the task is to predict the class label of a target image whether its label belongs to the *seen* or *unseen* category. Similar to ZSL, the learning setting assumes that all class-level semantic features are given, while only the images of seen classes are available for training. By exploring the correlation between image features and the corresponding semantic features, the main idea of the proposed approach is to enrich the semantic-to-visual (S2V) embeddings via a seamless fusion of adaptive and generative learning. To this end, we extend the semantic features of each class by supplementing image-adaptive attention so that the learned S2V embedding can account for not only inter-class but also intra-class variations. In addition, to break the limit of training with images only from seen classes, we design a generative scheme to simultaneously generate *virtual* class labels and their visual features by sampling and interpolating over seen counterparts. In inference, a testing image will give rise to two different S2V embeddings, seen and virtual. The former is used to decide whether the underlying label is of the unseen category or otherwise a specific seen class; the latter is to predict an unseen class label. To demonstrate the effectiveness of our method, we report state-of-the-art results on four standard GZSL datasets, including an ablation study of the proposed modules.

## 1 INTRODUCTION

Different from conventional learning tasks, zero-shot learning (ZSL) by Lampert et al. (2009); Palatucci et al. (2009); Akata et al. (2013) explores the extreme case of performing inference only over samples of unseen classes. To make the scenario more realistic, generalized zero-shot learning (GZSL) (Chao et al., 2016; Xian et al., 2017) is subsequently proposed so that inference can concern samples of both seen and unseen classes. Nevertheless. the learning setting in ZSL/GZSL is essentially the same where sample classes are divided into two categories, *seen* and *unseen*, but only those samples of seen classes are accessible to training. In addition, each of all the classes under consideration is characterized by semantic features such as attributes (Xian et al., 2018b) or text descriptions (Zhu et al., 2018) to specify and relate seen and unseen classes.

The lack of training samples from unseen classes has prompted generative approaches (Chen et al., 2018; Felix et al., 2018; Kumar Verma et al., 2018; Mishra et al., 2018) to creating synthetic data from semantic features of unseen classes. The strategy could enable learning semantic-visual alignment on unseen classes implicitly, and thus improves the ability to classify unseen classes. However, such generative models are indeed trained on seen samples, and the quality of synthesized unseen samples is predominantly influenced by seen classes. If the number of training samples of each seen class is small, it is hard for generative models to adequately synthesize samples of unseen classes, leading to unsatisfactory zero-shot learning. To better address the issue, we propose to synthesize visual and semantic features of *virtual* classes rather than those of the unseen classes. An interesting analogy is that childhood experience and relevant study (Greene, 1995) suggest the behavior of using human imagination to produce new object concepts could assist our cognitive capability. To mimic people utilizing imagination for exploring new knowledge, we create virtual classes by the integration of past "experience" (seen classes). In detail, we extend the *mixup* technique by Zhang et al. (2018) to

generate virtual classes, with a subtle difference that mixing is conducted on the semantic features (in addition to the visual ones), instead of the class label vectors.

In ZSL/GZSL, each seen or unseen class is typically described by a single semantic feature vector. The practice is useful in differentiating different classes in a principled way, but may not be sufficient to reflect the inter-class and intra-class visual discrepancies, not to mention the ambiguities caused by different backgrounds, view orientations, or occlusion in images. The concern of inefficient class-level representation can also be observed from how the semantic feature vectors are constructed. Take, for example, the Attribute Pascal and Yahoo (`aPY`) dataset (Farhadi et al., 2009), where each instance is annotated by 64 attributes. The semantic features of each class in `aPY` are obtained by averaging the attribute vectors of all its instances. We are thus motivated to introduce an image-adaptive class representation, integrating the original semantic features for inter-class discrimination with an image-specific attention vector for intra-class variations.

With the addition of virtual training data and the image-adaptive class representation, our method is designed to learn two classification experts: one for seen classes and the other for unseen classes. Both experts project the image-adaptive semantic feature vectors to the visual space and use cosine similarity to find the class label most similar to the given visual feature vector. The seen expert is trained with the provided training (seen) data, while the class prediction is over all possible classes, including seen and unseen. In inference, if its predicted class is not within the seen category. The testing sample is deemed to be from the unseen category, whose label is then decided by the unseen expert. The unseen expert is trained with the virtual data only, and the process indeed resembles meta-learning. However, the effectiveness of meta-learning is boosted by the design of the image-adaptive mechanism in that fine-tuning is not needed in performing zero-shot classification over unseen classes. We characterize the main contributions of this work as follows.

- Instead of generating synthetic data of unseen classes, we propose to yield virtual classes and data by mixup interpolations. The virtual classes of synthetic data can then be seamlessly coupled with meta-learning to improve the inference on unseen testing samples.
- We introduce the concept of representing each class with image-adaptive semantic features that could vary among intra-class samples. While the adaptive mechanism improves classifying the seen classes, it manifests the advantage in boosting the effect of meta-learning over virtual data to zero-shot inference over unseen classes.
- We demonstrate state-of-the-art results of zero-shot learning over four popular benchmark datasets and justify the design of our method with a thorough ablation study.

## 2 RELATED WORK

We review relevant literature in this section. First, we describe generative approaches for ZSL/GZSL that synthesize unseen images for training. To improve GZSL performance, we propose to couple virtual class generation with meta-learning for mimicking the inference scenario. Next, we discuss attention approaches that extract discriminating features from images to help classification.

### 2.1 GENERATIVE APPROACHES FOR ZSL/GZSL

Arguably one of the most important problems in ZSL/GZSL is to prevent models from being biased to seen classes. Generative approaches (Chen et al., 2018; Felix et al., 2018; Kumar Verma et al., 2018; Mishra et al., 2018; Schonfeld et al., 2019; Paul et al., 2019; Xian et al., 2019) tackle the problem by synthesizing visual features of unseen classes from their semantic features with generative models like Generative Adversarial Networks (GAN; Goodfellow et al., 2014) or Variational Autoencoders (VAE) (VAE; Kingma & Welling, 2014). The synthetic visual features act as pseudo-examples of unseen classes, and effectively reduce the ZSL/GZSL problem to a supervised learning one.

One key issue behind generative approaches comes from the insufficient amount of data to learn a good generative model. As a consequence, some semantic features that seem important during training may cause overfitting, and others that seem less important may be completely dropped. Therefore, several prior techniques propose new constraints or losses to preserve semantic features and regularize the generative model. For instance, (Chen et al., 2018) avoids the dropping of semantic information by disentangling the semantic space into two subspaces, one for classification and the

other for reconstruction; Felix et al. (2018) enforces visual-semantic feature consistency by requiring synthesized visual features to reconstruct semantic features accurately.

However, even with the new constraints or losses, the success of generative approaches still highly depends on whether there is enough data/variation in the seen classes to synthesize diverse visual features. Otherwise, the yielded features will be too close to those few seen classes and cannot help ZSL/GZSL much. For example, in `AWA2` dataset (Xian et al., 2018a), the unseen samples of "rat" are easily misclassified as seen classes of "mouse" or "hamster" because synthetic visual features for "rat" are doomed to be confused with seen visual features of those similar classes. The phenomenon inspires us to generate synthetic visual features for not only the given unseen classes but also the *virtual* unseen classes. These virtual classes provide a wider spectrum of support for unseen scenarios to improve ZSL/GZSL performance.

Our other focal effort is to connect the concept of meta-learning (Vinyals et al., 2016; Finn et al., 2017) with virtual classes in the training phase of the unseen expert. The most closely related approaches in ZSL are Li et al. (2019b); Yu et al. (2020); Verma et al. (2020); Sung et al. (2018); Hu et al. (2018). Verma et al. (2020) combines meta-learning and generative models to conquer limits in generative models. Li et al. (2019b) mimics the inference scenario by randomly selecting seen classes as "fake" new classes in each episode. Yu et al. (2020) randomly splits seen classes into two sets to train and refine the model. In contrast, we simulate the ZSL inference scenario in each episode by creating virtual classes from seen classes using mixup. Owing to the training scenario more resembling the inference setting in ZSL, our model achieves better S2V embedding on unseen classes and obtains state-of-the-art ZSL performance on most datasets. Besides, Sung et al. (2018) learns a deep distance metric to classify samples. Hu et al. (2018) utilizes the correction module to assist classification. On the contrary, our model does not need to learn complex relationship between classes and additional assistant module to achieve good performance.

## 2.2 ATTENTION

Attention mechanism is widely used in ZSL (Ji et al., 2018; Xie et al., 2019; Huynh & Elhamifar, 2020; Min et al., 2020; Liu et al., 2019). Highlighting important local features and reducing noisy feature influence generate a more effective mapping between visual and semantic domains. Ji et al. (2018) proposes $S^2GA$ which utilizes semantic features to emphasize most informative visual local features. (Xie et al., 2019) employs AREN and ACSE to focus on the most important region in images. Huynh & Elhamifar (2020) applies dense attribute-attention to find the most discriminating image parts and embed them to semantic features individually. DVBE (Min et al., 2020) utilizes spatial and channel attention to maximize inter-class margin. While the attention to visual features is proved useful by the above-mentioned research efforts, relatively few attempts assess the importance of attention on semantic features. In the ZSL setting, each class contains only one semantic feature vector such that intra-class variations are neglected from the single semantic representation. For example, the object of interest can be occluded partly in the image or in the front of different backgrounds, and a unique class-wise semantic feature vector is hard to reflect such variances. Thus, attention to semantic features is worth further exploring in ZSL. Along this line, the most relevant work to ours is LFGAA (Liu et al., 2019), which considers semantic prediction of samples. Different from LFGAA, our method puts attention on discriminating dimensions of semantic features and maps them to the visual domain. The effect of the proposed attention mechanism is to attract the visual representation of ground-truth class and repel others. Besides, our other main difference from LFGAA is the shortcut design in the attention model. We add a shortcut to prevent the model from overfitting on training data and retain discriminating dimensions for unseen class classification.

## 3 METHOD

We coin the proposed method as AGZSL in that its main idea is the fusion of Adaptive and Generative mechanisms in solving ZSL/GZSL. (See Figure 1.) We describe below the details of AGZSL.

### 3.1 NOTATIONS AND FORMULATION

We denote training data as $\mathcal{D} = \mathcal{D}_S \cup \mathcal{D}_U$ where $\mathcal{D}_S = \{(\mathbf{x}_s \in \mathcal{X}_s, \mathbf{a}_s \in \mathcal{A}_s, y_s \in \mathcal{Y}_s)\}$ comprises seen data and $\mathcal{D}_U = \{(\mathbf{a}_u \in \mathcal{A}_u, y_u \in \mathcal{Y}_u)\}$ includes unseen class labels $y_u$ and the corresponding

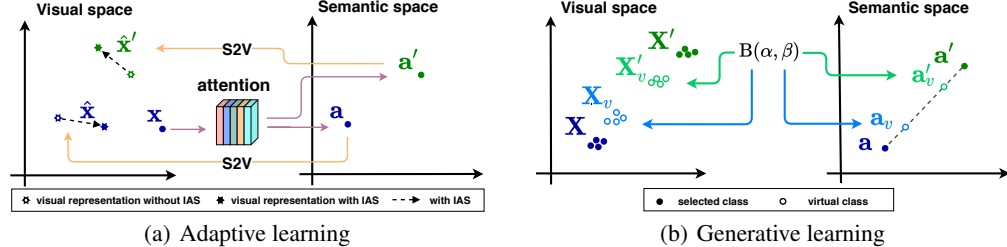

(a) Adaptive learning        (b) Generative learning

Figure 1: Overview of AGZSL. The same color reflects the same class association between visual features and attributes. (a) Image-adaptive mechanism (marked as purple) based on $\mathbf{x}$ yields proper attention to adapt the semantic features of $\mathbf{a}$ and $\mathbf{a}'$ so that their S2V embedding (marked as orange) displays the desired effects in $\hat{\mathbf{x}}$ (attracted to $\mathbf{x}$) and in $\hat{\mathbf{x}}'$ (repelled from $\mathbf{x}$). (b) We generate virtual classes from a pair of classes of semantic features $\mathbf{a}$ and $\mathbf{a}'$ with selected weights $\lambda$ via beta distribution $B(\alpha, \beta)$. $\mathbf{X}_v/\mathbf{X}'_v$ and $\mathbf{a}_v/\mathbf{a}'_v$ represent visual and semantic features of virtual classes.

semantic/attribute feature vectors $\mathbf{a}_u$. The dimensions of visual features and semantic features are respectively expressed as $\mathbf{x} \in \mathbb{R}^d$ and $\mathbf{a} \in \mathbb{R}^k$, while the number of total classes is $n$. We also assume that the seen category $\mathcal{Y}_s$ and the unseen category $\mathcal{Y}_u$ are disjoint, i.e., $\mathcal{Y}_s \cap \mathcal{Y}_u = \emptyset$ and $\mathcal{Y}_s \cup \mathcal{Y}_u = \mathcal{Y}$. Besides, we define $\mathcal{A}_s \cup \mathcal{A}_u = \mathcal{A} \in \mathbb{R}^{k \times n}$ as semantic features of all classes. Finally, the virtual data via mixup interpolations is denoted as $\mathcal{D}_V = \{(\mathbf{x}_v \in \mathcal{X}_v, \mathbf{a}_v \in \mathcal{A}_v, y_v \in \mathcal{Y}_v)\}$. As we explain later, $y_v$ is simply a new label of virtual class, rather than from label interpolation as in Zhang et al. (2018).

To perform the classification task, we explore the similarity correlations in the visual domain and consider semantic-to-visual (S2V) embeddings. Analogous to COSMO (Atzmon & Chechik, 2019), our method includes seen and unseen experts, $f_s$ and $f_u$, to classify testing samples as shown in Figure 2; however, the algorithmic designs are fundamentally different. The novel component of our method is the introduction of Image-Adaptive Semantics (IAS) in both experts to expand the spectrum of S2V embeddings. Specifically, in training, IAS diversifies the total mappings of S2V embedding from the number of different semantic vectors/classes to the number of samples. Its effect is twofold. First, it boosts the classification performance of the seen expert. Second, it generalizes the meta-learning of the unseen expert with virtual data and yields good performance in the inference of samples from unseen classes. Given a testing sample $\mathbf{x}$ in inference, we apply the two learned experts by

$$\hat{y}_s = \arg\max_{y \in \mathcal{Y}} f_s(\mathbf{x}, \mathcal{A}) \quad \text{and} \quad \hat{y}_u = \arg\max_{y \in \mathcal{Y}_u} f_u(\mathbf{x}, \mathcal{A}_u). \tag{1}$$

To decide the class label $\hat{y}$ of $\mathbf{x}$, we use the seen expert to discriminate whether the sample is from the seen category. If it is the case then $\hat{y}$ is decided by $f_s$, and otherwise by $f_u$. That is, we have

$$\hat{y} = \begin{cases} \hat{y}_s & \text{if } \hat{y}_s \in \mathcal{Y}_s, \\ \hat{y}_u & \text{otherwise.} \end{cases} \tag{2}$$

## 3.2   Image Adaptive Semantics (IAS)

As ZSL/GZSL assumes one semantic feature vector for each class, the conventional semantic-to-visual architecture leads to one single class-level visual feature vector, which is common to all samples of the same class. To learn such a single embedding vector with the constraint of being simultaneously similar to all visual features of samples from the same class tends to yield unsatisfactory classification outcomes in that it simply ignores the intra-class variations in visual features.

Diversifying semantic-to-visual embeddings with IAS establishes the core of our model as shown in Figure 3. To realize the concept, the network module learns to map image-adaptive semantic features to the corresponding visual features, according to the cosine similarity. To this effect, IAS leverages the given visual features $\mathbf{x}$ to focus on discriminative dimensions of the respective semantic features. Specifically, we apply IAS to obtain modified semantic features $\hat{\mathcal{A}}(\mathbf{x}) \in \mathbb{R}^{k \times n}$ by

$$\hat{\mathcal{A}}(x) = \mathcal{A} + \mathcal{A} \odot \text{softmax}(g(\mathbf{x})), \tag{3}$$

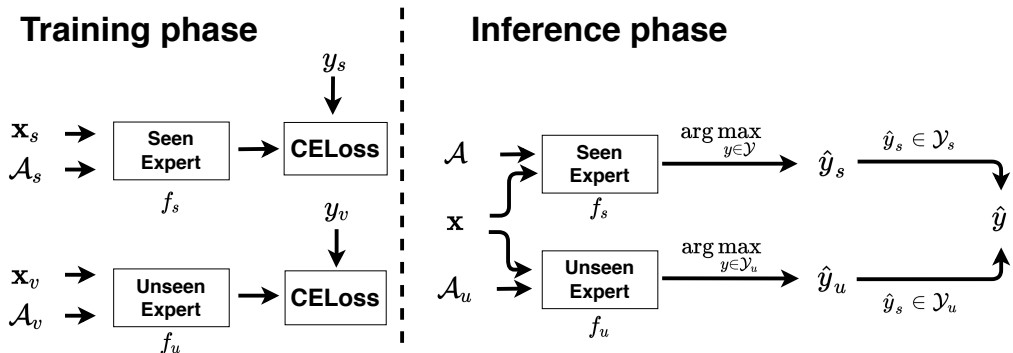

Figure 2: The proposed seen and unseen experts. In the learning phase, the seen expert $f_s$ is trained with seen visual features $\mathbf{x}_s$, seen semantic features $\mathcal{A}_s$, and corresponding class labels $y_s$, while the unseen expert $f_u$ is trained via meta-learning with virtual data. In the inference phase, given a testing sample $\mathbf{x}$, $f_s$ is additionally fed with all semantic features $\mathcal{A}$, and $f_u$ only with unseen semantic features $\mathcal{A}_u$. The decision rule leading to the class label $\hat{y}$ is based on (2). Note that in both training and inference stages, the image-adaptive mechanism to semantic features is carried out.

where $\text{softmax}(g(\mathbf{x})) \in \mathbb{R}^k$ represents using a linear layer $g$ to predict semantic attention and $\odot$ denotes Hadamard product to combine attention and semantic features. Note that the shortcut implementation in (3) is pivotal as the softmax function often puts high attention weights on sparse dimensions and draws others near to zero. Without the shortcut, the modified $\hat{\mathcal{A}}$ would miss some part of semantic features and degenerate the classification performance. Also, it is worthy to point out that compared to LFGAA (Liu et al., 2019), our method pays attention to the semantic features of classes, not the semantic prediction of the model. With (3), the process to drive the image-adaptive class-level visual features $\mathcal{V}(\mathbf{x})$ can be expressed by

$$\mathcal{A} \in \mathbb{R}^{k \times n} \xrightarrow[\text{IAS}]{\mathbf{x} \in \mathbb{R}^d} \hat{\mathcal{A}}(\mathbf{x}) \in \mathbb{R}^{k \times n} \xrightarrow[\text{S2V}]{} \mathcal{V}(\mathbf{x}) \in \mathbb{R}^{d \times n}. \qquad (4)$$

Once we have obtained the class-level visual features, we then use the cosine similarity and softmax to calculate the probability of each class. We have the class probability of prediction on $\mathbf{x}$:

$$\mathbf{p}(\mathbf{x}) = f(\mathbf{x}, \mathcal{A}) = \text{softmax}(\sigma \times \cos(\mathcal{V}(\mathbf{x}), \mathbf{x})), \qquad (5)$$

where $\sigma$ is the learnable scale and $f$ is indeed the general form of a classification expert. We conclude by stating that the Cross-Entropy(CE) loss is adopted to update the model:

$$\mathcal{L}_{\text{CE}} = -\sum_{\mathbf{x}} \log p_y(\mathbf{x}), \qquad (6)$$

where $p_y(\mathbf{x})$ is the predicted probability of ground-truth label.

### 3.3 SEEN EXPERT

The objective of seen expert $f_s$ is to recognize seen classes and otherwise decide whether a testing sample belongs to unseen classes in inference phase. In zero-shot learning, especially GZSL, the capability to single out testing samples of unseen classes is crucial to the classification performance. If one can establish reasonable split mechanisms such as in Min et al. (2020) and Chen et al. (2020), the more challenging GZSL would be simplified into ZSL. For example, COSMO (Atzmon & Chechik, 2019) utilizes Confidence Base Gating to discriminate between seen and unseen classes, while DVBE (Min et al., 2020) considers the entropy threshold. In our formulation, the seen expert learns to predict all possible classes, including seen and unseen, although the training data assume only seen class labels. That is, $f_s$ is trained to map (image adaptive) seen semantic features to their corresponding visual features. Such optimization can be thought of as one-category anomaly detection over only seen classes. Thus, in inference, if the highest cosine similarity value by the seen expert $f_s$ is not yielded by one of the seen classes, the testing sample is considered *abnormal* from the unseen category and its classification should be decided by the unseen expert discussed next.

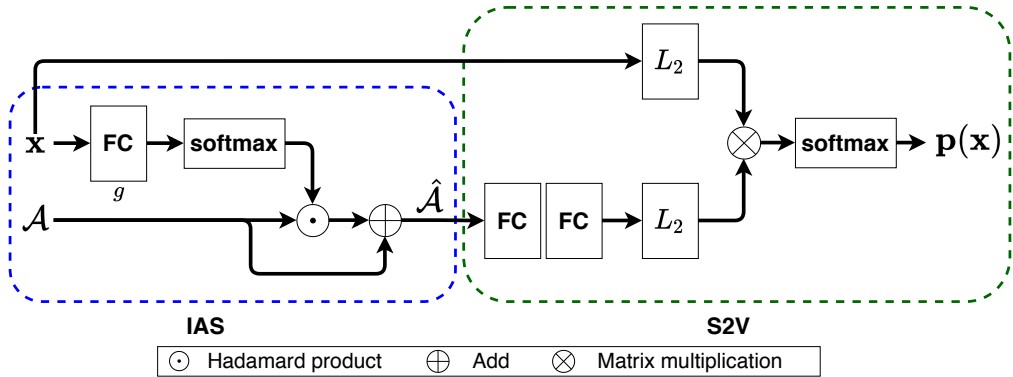

Figure 3: The architectures of IAS and S2V (FC: linear layer, $L_2$: $L_2$-normalization).

### 3.4 UNSEEN EXPERT

So far we have stated the objective of the unseen expert $f_u$ is to classify the labels of those that cannot be decided by the seen expert, as in (2). To resolve the dilemma of zero-shot setting, we apply meta-learning to train $f_u$ over virtual classes by mixup interpolations. The exact number of virtual classes in each episode is decided by the pilot study (Li et al., 2019b), and an ablation study in appendix is provided to analyze its effect on the performance of zero-shot classification. To begin with, in each episode we randomly select $m$ pairs of seen classes. Then, we create $m$ virtual classes out of these pairs by mixup. The semantic and visual features for these virtual classes are defined as

$$\mathbf{a}_v = \lambda \, \mathbf{a}_i + (1 - \lambda) \, \mathbf{a}_j, \tag{7}$$
$$\mathbf{x}_v = \lambda \, \mathbf{x}_i + (1 - \lambda) \, \mathbf{x}_j, \tag{8}$$

where $\mathbf{a}_i$ and $\mathbf{a}_j$ are semantic features of paired classes, and $\mathbf{x}_i$ and $\mathbf{x}_j$ are randomly chosen from visual samples of the paired classes. Following Zhang et al. (2018), $\lambda \in [0, 1]$ is sampled from the beta distribution. We then define a new virtual class $y_v$, with $\mathbf{a}_v$ as semantic features, to be associated with virtual visual features $\mathbf{x}_v$. Previous generative models (Schonfeld et al., 2019; Xian et al., 2019) synthesize unseen samples by GAN/VAE or their variants. In comparison, the proposed meta-learning has at least three advantages for ZSL. First, the small number of classes in some datasets, such as 50 in AWA2, may fall short to provide sufficient support set such that the generative model is hard to be optimized, while episodic meta-learning over virtual classes can overcome this problem. Second, some part of semantic features which are important to classify unseen classes do not have enough training classes in the datasets. With mixup semantic features as virtual classes, our model can learn underlying discriminating semantic features well. Third, the fusion of image-adaptive mechanism and meta-learning is flexible. As a result, we do not need online fine-tuning in applying the unseen expert $f_u$ yielded by meta-learning to predict the label of unseen class for testing samples.

## 4 EXPERIMENT

### 4.1 EXPERIMENTAL SETTING

**Datasets.** In GZSL, there are four widely used benchmark datasets for evaluation, which are AWA2 (Xian et al., 2018a), USCD Birds-200-2011 (CUB) (Welinder et al., 2010), SUN (Patterson & Hays, 2012), and aPY (Farhadi et al., 2009). AWA2 contains 37K images and 50 kinds of animals with 85-dimension attributes. CUB consists of 11K images and 200 bird species described with 312-dimension attributes. SUN is composed of 14K images 717 classes labeled with 102-dimension attributes. aPY includes 15K images and 32 different classes with 64-dimension attributes. The attributes of class in datasets will be taken as the semantic features in our work. We follow existing GZSL approaches (Min et al., 2020; Li et al., 2019b) and extract 2048-dimensional visual features for each image by the ResNet101 backbone (He et al., 2016) pretrained on ImageNet-1K. We then split the data into seen and unseen classes according to the benchmark procedure from Xian et al. (2017).

Table 1: GZSL results on four datasets. All methods in comparison utilize ResNet101 as the backbone for fairness. Notation "*" means the method fine-tunes the backbone to match the characteristic of datasets.

| | AWA2 | | | | CUB | | | | SUN | | | | aPY | | | |
|---|---|---|---|---|---|---|---|---|---|---|---|---|---|---|---|---|
| | ZSL | GZSL | | | ZSL | GZSL | | | ZSL | GZSL | | | ZSL | GZSL | | |
| | T1 | U | S | H | T1 | U | S | H | T1 | U | S | H | T1 | U | S | H |
| SP-AEN | 58.5 | 23.3 | 90.9 | 37.1 | 55.4 | 34.7 | 70.6 | 46.6 | 59.2 | 24.9 | 38.6 | 30.3 | 24.1 | 13.7 | 63.4 | 22.6 |
| DLFZRL | 60.9 | - | - | 45.1 | 51.9 | - | - | 37.1 | 42.5 | - | - | 24.6 | 38.5 | - | - | 31.0 |
| PSR | 63.8 | 20.7 | 73.8 | 32.3 | 56.0 | 24.6 | 54.3 | 33.9 | 61.4 | 20.8 | 37.2 | 26.7 | 38.4 | 13.5 | 51.4 | 21.4 |
| CDL | - | - | - | - | 54.5 | 23.5 | 55.2 | 32.9 | 63.6 | 21.5 | 34.7 | 26.5 | 43.0 | 19.8 | 48.6 | 27.1 |
| PQZSL | - | - | - | - | - | 53.2 | 51.4 | 46.9 | - | 35.1 | 35.3 | 35.2 | 27.9 | 64.1 | 64.1 | 38.8 |
| f-VAEGAN-D2* | 70.3 | 57.1 | 76.1 | 65.2 | 72.9 | 63.2 | 75.6 | 68.9 | 65.6 | 50.1 | 37.8 | 43.1 | - | - | - | - |
| LsrGAN | - | 54.6 | 74.6 | 63.0 | - | 48.1 | 59.1 | 53.0 | - | 44.8 | 37.7 | 40.9 | - | - | - | - |
| TF-VAEGAN* | 73.4 | 55.5 | 83.6 | 66.7 | 74.3 | 63.8 | 79.3 | 70.7 | 66.7 | 41.8 | 51.9 | 46.3 | - | - | - | - |
| OCD-CVAE | 71.3 | 59.5 | 73.4 | 65.7 | 60.9 | 44.8 | 59.9 | 51.3 | 62.1 | 44.8 | 42.9 | 43.8 | - | - | - | - |
| ZSML Softmax | 76.1 | 58.9 | 74.6 | 65.8 | 69.6 | 60.0 | 52.1 | 55.7 | 60.2 | - | - | - | 64.1 | 36.3 | 46.6 | 40.9 |
| GXE | 71.1 | 56.4 | 81.4 | 66.7 | 54.4 | 47.4 | 47.6 | 47.5 | 62.6 | 36.3 | 42.8 | 39.3 | 38.0 | 26.5 | 74.0 | 39.0 |
| E-PGN | 73.4 | 52.6 | 83.5 | 64.6 | 72.4 | 52.0 | 61.1 | 56.2 | - | - | - | - | - | - | - | - |
| Relation Net | 64.2 | 30.0 | 93.4 | 45.3 | 55.6 | 38.1 | 61.1 | 47.0 | - | - | - | - | - | - | - | - |
| Correlation Net | - | - | - | - | 45.8 | 41.9 | - | - | - | - | - | - | - | - | - | - |
| LFGAA+Hibrid | 68.1 | 27.0 | 93.4 | 41.9 | 67.6 | 36.2 | 80.9 | 50.0 | 62.0 | 18.5 | 40.0 | 25.3 | - | - | - | - |
| AREN* | 66.9 | 54.7 | 79.1 | 64.7 | 72.5 | 63.2 | 69.0 | 66.0 | 60.6 | 40.3 | 32.3 | 35.9 | 39.2 | 30.0 | 47.9 | 36.9 |
| COSMO | - | - | - | - | - | 44.4 | 57.8 | 50.2 | - | 44.9 | 37.7 | 41.0 | - | - | - | - |
| DVBE* | - | 62.7 | 77.5 | 69.4 | - | 64.4 | 73.2 | 68.5 | - | 44.1 | 41.6 | 42.8 | - | 37.9 | 55.9 | 45.2 |
| ours | 73.8 | 65.1 | 78.9 | 71.3 | 57.2 | 41.4 | 49.7 | 45.2 | 63.3 | 29.9 | 40.2 | 34.3 | 41.0 | 35.1 | 65.5 | **45.7** |
| ours* | **76.4** | 69.0 | 86.5 | **76.8** | **77.2** | 69.2 | 76.4 | **72.6** | 66.2 | 50.5 | 43.1 | **46.5** | 43.7 | 36.2 | 58.6 | 44.8 |

**Evaluation.** We evaluate our approach for both ZSL and GZSL settings. For the ZSL setting, the average per-class top-1 accuracy (**T**) on unseen classes is taken as the evaluation metric. For GZSL (Xian et al., 2017), the evaluation metrics include seen classes accuracy **S**, unseen class accuracy **U**, and their harmonic mean $\mathbf{H} = (2\mathbf{U} \cdot \mathbf{S})/(\mathbf{U} + \mathbf{S})$.

**Implementation.** As suggested in Li et al. (2019b), we normalize the visual and semantic features into $[0, 1]$. The architecture of semantic-to-visual embedding contains a two-layer linear model with 1,600 hidden units and utilizes ReLU on the hidden and output layer. The seen and unseen experts are trained by Adam optimizer with a learning rate $5 \times 10^{-5}$ and $5 \times 10^{-4}$ respectively for all datasets. We apply 200,000 episodes to train the unseen expert. In each episode, we randomly generate 16 or 20 (based on the dataset) virtual classes and 4 samples for each class. We follow the setting in (Min et al., 2020) to fine-tune the backbone.

## 4.2 COMPARISON WITH STATE-OF-THE-ART APPROACHES

We compare our method with 13 recent GZSL approaches. These include SP-AEN (Chen et al., 2018), DLFZRL (Tong et al., 2019), PSR (Annadani & Biswas, 2018), CDL (Jiang et al., 2018), PQZSL (Li et al., 2019a), f-VAEGAM-D2 (Xian et al., 2019), LsrGAN (Vyas et al., 2020), TF-VAEGAN (Narayan et al., 2020), OCD-CVAE (Keshari et al., 2020), ZSML Softmax (Verma et al., 2020), E-PGN (Yu et al., 2020), Relation Net (Sung et al., 2018), Correlation Net (Hu et al., 2018), GXE Li et al. (2019b), LFGAA+Hybrid (Liu et al., 2019), AREN (Xie et al., 2019), COSMO (Atzmon & Chechik, 2019), DVBE (Min et al., 2020). Table 1 shows that the proposed AGZSL achieves the best GZSL performance on all four datasets. It improves harmonic mean by 7.4% on AWA2, 1.9% on CUB, 0.2% on SUN, and 0.5% on aPY, respectively. Particularly, among those methods in comparison, DVBE needs to search the entropy threshold $\tau$ and the embedding model for the best harmonic means according to each dataset. In contrast to DVBE, without entropy threshold and specific model searching, our model achieves over 4% improvement on most datasets. The main reason for better performance is that AGZSL provides more discriminating visual representation for both seen and unseen classes. Therefore, our model can effectively separate seen and unseen samples and achieve better performance in the harmonic mean. Notably, AGZSL drops performance on aPY after backbone fine-tuning. It could be due to that its number of seen classes and training images are fewest among the four datasets. Fine-tuning the backbone causes the visual features to overfit training samples and become less discriminating for correct classification.

Table 2: The effect of virtual classes and IAS for GZSL. We remove these two components as our baseline (b). In the ablation study, we add virtual classes (v) and IAS (I) step by step to show their effect on GZSL.

| | AWA2 | | | | CUB | | | | SUN | | | | aPY | | | |
|---|---|---|---|---|---|---|---|---|---|---|---|---|---|---|---|---|
| | ZSL | GZSL | | | ZSL | GZSL | | | ZSL | GZSL | | | ZSL | GZSL | | |
| | T1 | U | S | H | T1 | U | S | H | T1 | U | S | H | T1 | U | S | H |
| b | 72.2 | 63.2 | 86.1 | 72.9 | 73.7 | 68.6 | 71.3 | 69.9 | 63.9 | 47.6 | 42.1 | 44.7 | 40.0 | 33.4 | 57.2 | 42.2 |
| b+v | 75.8 | 66.9 | 86.1 | 75.3 | 76.1 | 70.3 | 71.3 | 70.8 | 65.6 | 48.6 | 42.1 | 45.1 | 42.8 | 35.2 | 57.2 | 43.6 |
| b+v+I | **76.4** | 69.0 | 86.5 | **76.8** | **77.2** | 69.2 | 76.4 | **72.6** | **66.2** | 50.5 | 43.1 | **46.5** | **43.7** | 36.2 | 58.6 | **44.8** |

Furthermore, generative models achieve higher performance on CUB and SUN, because in these two datasets there are many seen classes for synthesizing unseen samples of better quality. In comparison, our model overcomes the influence of few seen classes and also achieves the best performance on AWA2 and aPY. Our results empirically support that the introduction of virtual classes and samples are indeed helpful to classify unseen classes. Finally, we remark that since the backbone is pretrained on ImageNet-1K, which is rather different from the ZSL fine-grained databases, e.g., CUB and SUN. Thus, the harmonic mean improves a lot after fine-tuning the backbone on these two datasets.

### 4.3 ABLATION STUDY

To evaluate the benefits of tackling ZSL/GZSL with virtual classes and IAS, we carry out an ablation study on the four datasets. We incrementally include each key component to assess their effect. As shown in Table 2, the virtual classes improve ZSL accuracy significantly. The ZSL performance improves 4.6% on AWA2, 2.4% on CUB, 1.7% on SUN, and 2.8% on aPY, respectively. The result implies that the virtual classes are advantageous to ZSL. After adding IAS to our model, both seen and unseen accuracy increases, and the harmonic mean H improves about 1.5% on average. The consistent gain exemplifies that IAS can provide better semantic-to-visual embedding and achieve higher performance on GZSL.

### 4.4 COSINE SIMILARITY MARGIN

Figure 4 illustrates the effect of the proposed AGZSL model. The cosine similarity margin means the inner product value between the correct class and the other nearest class. A negative margin indicates that the model predicts a wrong class. On the other hand, a large positive margin implies that the model results in a proper embedding distinguishing the correct class from others effectively. Thus, seeking a positive and larger margin is a crucial and important objective in GZSL. As shown in the first row of Figure 4, the margins of unseen samples shift right on all four datasets by our method. It shows that AGZSL indeed generates better embedding on the visual space and makes the model easier to predict the correct class. Moreover, from the second row in Figure 4, the plots show that our method improves the margins of seen samples dramatically on most datasets. The above experiment explains that AGZSL can achieve a higher margin for most samples and achieves better classification performance on the four testing datasets.

### 4.5 THE DISTRIBUTION OF VIRTUAL CLASSES

To explicitly exhibit the distributions of seen, unseen and virtual classes, we apply PCA to map the class-wise semantic features to a two-dimensional space, as shown in Figure 5. The features of virtual classes scatter around seen classes, because we have used $\mathbf{B}(5,1)$ to generate virtual classes. Advantageously, some virtual classes are very close to unseen classes in the semantic domain and turn out to be helpful for enabling the unseen expert to classify samples of unseen classes. With meta-learning over these numerous virtual classes and samples, the proposed S2V can better learn the embedding between semantic and visual domains, and results in an effective unseen expert.

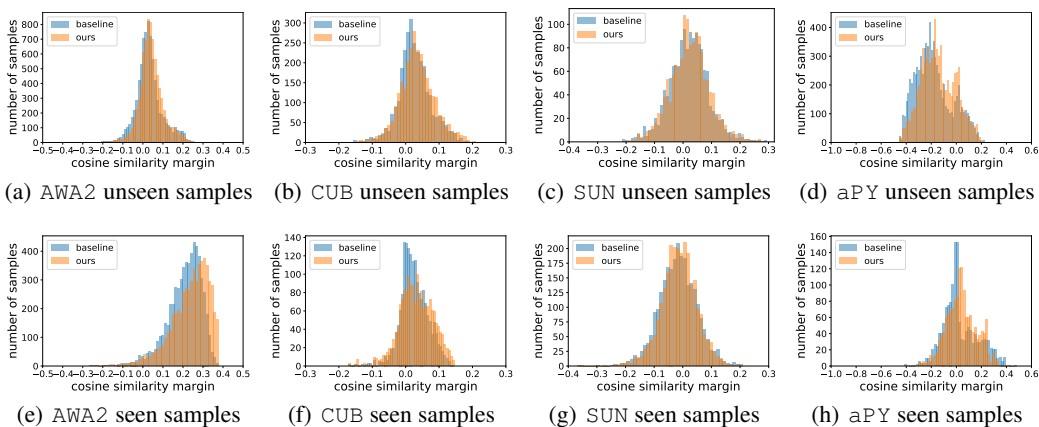

(a) AWA2 unseen samples    (b) CUB unseen samples    (c) SUN unseen samples    (d) aPY unseen samples

(e) AWA2 seen samples    (f) CUB seen samples    (g) SUN seen samples    (h) aPY seen samples

Figure 4: The cosine similarity margin between the correct class and other nearest class. The blue area depicts the cosine similarity margin of baseline without IAS and generative learning. On the other hand, the orange area represents the cosine similarity margin of ours. Most of the samples improve their margin by our method, suggesting the proposed techniques are beneficial for GZSL.

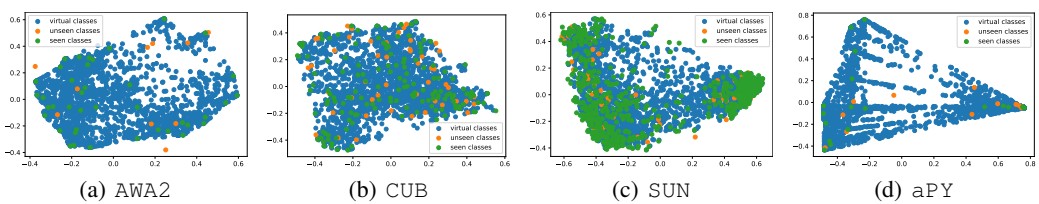

(a) AWA2    (b) CUB    (c) SUN    (d) aPY

Figure 5: The semantic features of virtual/seen/unseen classes over the four datasets.

## 5 CONCLUSION

In this paper, we propose Image Adaptive Semantics (IAS) and meta-learning with virtual classes and samples to solve the ZSL/GZSL problem. To deal with the intra-class visual discrepancies, IAS adaptively emphasizes the most discriminating dimensions in semantic features with respect to the underlying visual features. To better classify samples of the unseen classes in inference, we propose a novel formulation of generative meta-learning. Different from previous generative models that focus on synthesizing unseen samples for training the model, we create virtual classes and their respective virtual samples in the training phase. Further, to imitate the ZSL inference scenario, we carry out meta-learning with these virtual data to extend our model. In our experiment, we demonstrate that AGZSL is beneficial to tackle the challenging ZSL/GZSL problem and achieves significant advantages over those in comparison on the four ZSL/GZSL benchmark datasets.

## ACKNOWLEDGEMENTS

This work was supported in part by the MOST, Taiwan under Grants 110-2634-F-001-009 and 107-2628-E-002-008-MY3. We are grateful to the National Center for High-performance Computing for providing computational resources and facilities.

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

# A APPENDIX

Table 3: The impact of shortcut in IAS on four datasets.

| shortcut | AWA2 ZSL T | GZSL U | S | H | CUB ZSL T | GZSL U | S | H | SUN ZSL T | GZSL U | S | H | aPY ZSL T | GZSL U | S | H |
|---|---|---|---|---|---|---|---|---|---|---|---|---|---|---|---|---|
| | 23.9 | 13.0 | 30.0 | 18.2 | 23.3 | 17.0 | 45.8 | 24.8 | 42.1 | 22.5 | 20.3 | 21.4 | 30.5 | 14.9 | 4.7 | 7.1 |
| √ | **76.4** | 69.0 | 86.5 | **76.8** | **77.2** | 69.2 | 76.4 | **72.6** | **66.2** | 50.5 | 43.1 | **46.5** | **43.7** | 36.2 | 58.6 | **44.8** |

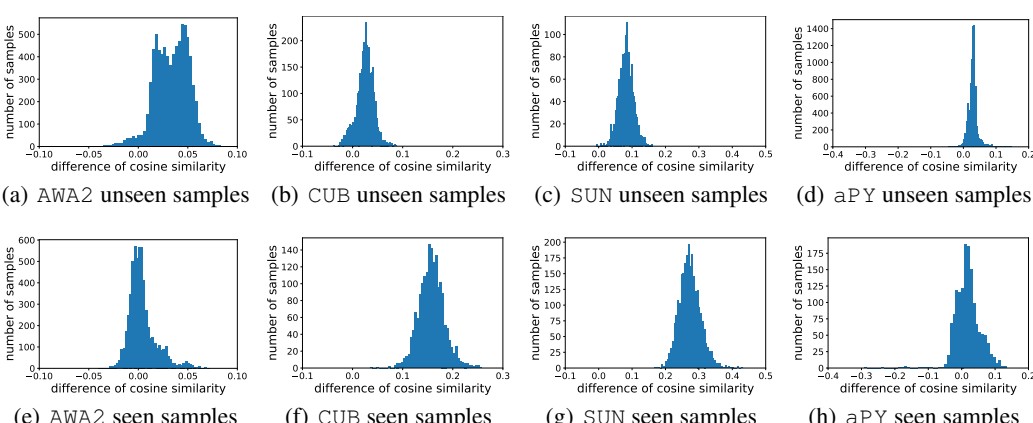

(a) AWA2 unseen samples  (b) CUB unseen samples  (c) SUN unseen samples  (d) aPY unseen samples

(e) AWA2 seen samples  (f) CUB seen samples  (g) SUN seen samples  (h) aPY seen samples

Figure 6: The difference of cosine similarity between our model and baseline. In our experiment, most of the samples improve their embedding with IAS and virtual classes.

**Shortcut in IAS**   Table 3 shows that shortcut in IAS improves performance a lot. The unseen class accuracy increase around 50% on AWA2 and SUN. Besides, the harmonic mean also enhances 20% to 60% on four datasets with the shortcut. The main reason is that the softmax usually puts high attention weight on few dimensions and decreases others near to zero. Thus, without shortcut, the model is prone to overfitting on training data.

**Cosine similarity enhancement**   Figure 6 illustrates better embedding for seen and unseen classes in our model. Since we apply cosine similarity to classify samples, the higher score means visual representation of each class is closer to their visual samples. The first row in 6 shows effect of our model on unseen classes. With virtual classes and IAS, cosine similarity increase averaged 0.05 to 0.1 on different datasets. The increasing amount shows our model provides better visual-to-semantic embedding for unseen classes and samples are easier to classify correctly. Therefore, our model can achieve good ZSL performance. Besides, the second row in 6 illustrates the effect of IAS on seen classes. Most of the testing seen samples increase their cosine similarity around 0.02 on four datasets. With the IAS and virtual classes, the model can achieve better visual-to-semantic embedding on all classes and obtain better the generalized seen accuracy on four datasets.

**virtual classes number per episode**   To search the class number in each episode for best ZSL, we search 4 different numbers, i.e, 4,8,12, and 20 per episode. Table 4 shows that the best class number setting is related to the class number of classes. For example, the AWA2 and aPY which include fewer unseen classes obtain the best ZSL accuracy with 16 virtual classes. On the other hand, the CUB and SUN which include more unseen classes achieve the best performance with 20 virtual classes. The reason is that the advantage of meta-learning is to simulate inference scenarios while training (Li et al., 2019b). The virtual class number set closer to the unseen class number is favorable in the meta-learning setting. Therefore, the best virtual class number is different according to the unseen number of the dataset.

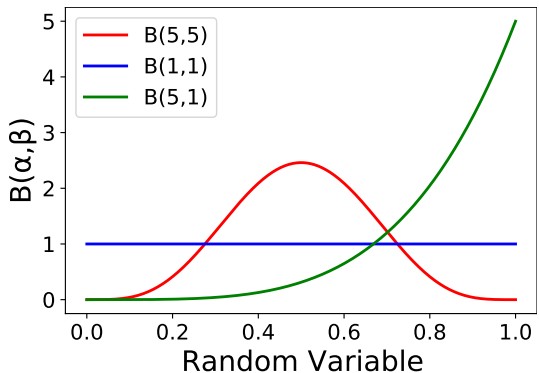

Figure 7: Beta distribution.

Table 4: ZSL accuracy w.r.t. virtual classes per episode.

| k | AWA2 | CUB | SUN | aPY |
|---|---|---|---|---|
| 8 | 75.7 | 76.2 | 64.6 | 40.8 |
| 12 | 76.0 | 76.3 | 65.1 | 42.8 |
| 16 | **76.4** | 76.3 | 65.1 | **43.7** |
| 20 | 75.5 | **77.2** | **66.2** | 41.1 |

Table 5: ZSL accuracy w.r.t. beta distribution.

| B($\alpha$,$\beta$) | AWA2 | CUB | SUN | aPY |
|---|---|---|---|---|
| (5,5) | 74.8 | 75.6 | 65.4 | 38.2 |
| (1,1) | 75.3 | 76.4 | 65.8 | 40.2 |
| (5,1) | **76.4** | **77.2** | **66.2** | **43.7** |

**Effect of beta distribution on ZSL accuracy**     To explore the best beta distribution setting for ZSL, our experiments apply three different $\alpha$ and $\beta$ sets. As shown in Figure 7, B(1,1) means virtual classes distribute uniformly between selected class pairs. Then, B(5,5) means most of the virtual classes distribute among the middle of selected class pairs. On the other hand, B(5,1) means most virtual classes distribute closer to one of the classes in selected class pairs. Table 5 demonstrates that B(5,1) achieves best performance on each dataset. In our speculation, the virtual classes created based on one class are closer to real-world classes. For example, the zebra is very similar to the horse in shape. Thus, the virtual classes distributing among seen classes enhance ZSL accuracy are very beneficial for training ZSL.

