# OpenReview forum: "Adaptive and Generative Zero-Shot Learning"
_ICLR.cc/2021/Conference — ICLR 2021 Poster_

### Official Review · AnonReviewer4 · 2020-10-26
**An interesting approach for zero-shot learning leveraging the instance attention and synthetic data to enhance the unseen classifier**

**Rating:** 5
**Confidence:** 5

**Review:**

Summary: The authors proposed an interesting method for zero-shot learning. In particular, the authors adopted an attention mechanism from the input feature in the semantic to visual mapping, to introduce intra-class variations in the visual space. They also propose a process to synthesize "fake" class representations such that a classifier for unseen classes can be trained.  Combining these two the authors demonstrated significant results on benchmark zero-shot learning datasets.

Reason for score: I think this paper is marginally below the acceptance threshold. It proposed an elegant method which produces state-of-the-art results on benchmark datasets. But more details need to be revealed to help readers understand the method (discussed below), and I hope the authors can address my concerns during the rebuttal.

Pros:
1. The paper proposed an interesting attention-based method to introduce intra-class variations to the representations for each class.

2. The authors proposed to synthesize virtual classes to enrich the data for training an unseen classifier.

3. The extensive experiments on the benchmark datasets illustrate the superiority of the proposed method.

Cons:
1. The overall training scheme is not very clear and I hope the authors can clarify them. For example, what is the order of training the seen/unseen experts? Do they share any component, like the IAS or S2V module?

2. Baseline explanation is missing. In Sec. 4.3, the authors demonstrate the ablation study where the baseline model does not have IAS or the virtual classes training scheme. How is this baseline implemented?  Without the virtual classes how would you train the unseen expert? My understanding is that without the IAS the model reduces to pure S2V and without the virtual classes the model can only train on seen classes, in which the entire model reduce to a single cosine similarity classifier in the visual feature space (given the seen features and semantics). Then how do you leverage the seen/unseen expert inference strategy? I am curious about it because this baseline reaches considerably high performance in Table 2, in fact already outperforms many state-of-the-art methods in Table 1. It is kind of shocking to me that such a naive baseline contributes this level of accuracy improvement, if I didn't misunderstand it. I would be appreciated if the authors can deliberate this part, so readers can understand the contribution of each module in the proposed method.

3. Why is seen expert detecting the abnormal? In Sec. 3.3, the authors claim that "Such optimization (the cross-entropy) can be thought of as one-category anomaly detection over only seen classes". During training, since the seen expert has no data for unseen classes, the classifier is easily biased to the seen classes, even the new data is from an unseen class in the inference time. That's one of the reasons why many embedding based zsl methods have very poor generalized zero-shot learning performance [Xian et al, cvpr 2017]. Could authors deliberate more on why this specific optimization (the cross-entropy on the seen) can address this issue, such that the model can work as a anomaly detector and predict higher score for unseen classes when unseen example is input? It would also be helpful if the authors can demonstrate the predicted scores of the unseen examples empirically.

---

> ### Author Response · Authors · 2020-11-24
> **Responses to AnonReviewer4**
>
> 1. The seen expert f_s and unseen expert f_u are two different models, which do not share any parameters. They are trained independently, where the seen expert is trained with the real data and the unseen expert is meta-trained with the virtual data. The architecture of such an expert is illustrated in Figure 3.
>
>
> 2. We will improve our writing to better explain the implementation of the baseline in Section 4.3. For the ablation study, the baseline also includes the seen and unseen experts. They do not include IAS and both use the image features from the fine-tuned ResNet101 backbone. So, learning the unseen expert of the baseline is now reduced to learning the S2V embedding for the unseen classes. We adopt the conventional zero-shot learning technique that employs episodic meta-learning over the seen classes/samples to obtain the zero-shot S2V embedding.
>
>
> 3. Although the seen expert is trained only with the samples from the seen classes, its class prediction is indeed over all classes (including both seen and unseen), as indicated in equation (1). In learning the seen expert, we expect most of the training samples from the various seen classes can be correctly predicted. Under this assumption, if a training sample is predicted by the seen expert not within any of the seen classes, it would be considered abnormal. In other words, we do not assume that a sample of an unseen class would yield high probabilities of unseen classes by the seen expert. After all, we do not have access to those data of unseen classes in training. In inference, if a sample is predicted by the seen expert as abnormal (i.e., not within the seen category), either it is from a seen class but cannot be correctly predicted by the seen expert, or it is from an unseen class. Either way, its classification will be decided by the unseen expert.

---

### Official Review · AnonReviewer1 · 2020-10-26
**Please find below for the detailed comments**

**Rating:** 7
**Confidence:** 4

**Review:**

This paper solves ZSL and GZSL problem by an elegant fusion of adaptive and generative learning. Different from previous generative models that synthesize unseen samples for training the model, authors create virtual classes as unseen classes in the training phase. In addition, four standard GZSL datasets demonstrate the effectiveness of the proposed method.
Strengths:
1. The motivation and the contribution are clearly presented in this manuscript. In general, this paper is well-written.
2. Instead of generating synthetic data of unseen classes, authors propose to yield virtual classes and data by mixup interpolations.
3. The architectures of IAS and S2V are simple and effective in terms of experimental results.
Weaknesses:
1. Parameter settings for the network are unclear.
2. More generative ZSL methods from CVPR’20 or ECCV’20 should be compared.
3. More experimental results such as a tsne illustration for generated virtual classes should be given.

---

> ### Author Response · Authors · 2020-11-24
> **Responses to AnonReviewer1**
>
> 4. We have made our code available on github, https://github.com/anonmous529/AGZSL.
> As shown in Figure 3, IAS is a single fc layer which transfers 2048-D image features to semantic features and utilizes ReLU on the output. The L_2 in Figure 3 means L_2 normalization. The network architecture of S2V embedding is described in the Implementation of Section 4.1.
> For learning the unseen expert via meta-learning, we apply 200 epochs and 1000 episodes for each epoch. In each episode we randomly generated 16 ~ 20 virtual classes and 4 samples for each virtual class. The unseen expert is trained by Adam optimizer with a learning rate 5*10^-5 for all datasets.
>
>
> 5. Thank you for the suggestion. We have found the following three papers on generative ZSL in CVPR'20 and ECCV'20. Among them, [b] is transductive learning which is different from our setting and [c] is already included in our comparisons. In the revised paper, we have included [a] in the comparisons as in Table 1 of Section 4.
>
>     [a] Generalized Zero-Shot Learning Via Over-Complete Distribution, cvpr2020.
>
>     [b] Self-supervised Domain-aware Generative Network for Generalized Zero-shot, cvpr2020.
>
>    [c] Latent Embedding Feedback and Discriminative Features for Zero-Shot Classification, eccv2020.
>
>
> 6. Thank you for the suggestion. In the revised paper, we have added a new section, namely, Section 4.5, to show the PCA distributions of the virtual classes.

---

### Official Review · AnonReviewer2 · 2020-10-27
**Review of "Adaptive and Generative Zero-Shot Learning"**

**Rating:** 6
**Confidence:** 3

**Review:**

I. Summary

The authors consider (generative) zero-shot classification. Their approach, combines two main aspects: (1) generating "virtual" classes by mixup interpolation and (2) they introduce an attention mechanism, dubbed "image-attentive attention" to allow their approach to model both intra- and extra- class variations.

II. Strong and weak points.

Positive:
- Strong results: overall the results are good. Yes, the paper may not account for the latest crop of papers from e.g. CVPR but I feel that is besides the point as these methods are structurally different.
- Relatively extensive evaluation: there is an ablation study, and a distinction over whether the image encoder is fine-tuned.
- Writing is clear, paper is reasonably easy to follow.

Negative:
- A large part of the improvement stems from the virtual classes, which is essentially mixup applied to ZSL. This is somewhat derivative.

III. Rating and questions to authors
Despite the negative comment I mentioned, I feel the paper should be accepted. How does the paper work with actual generated samples as in the GAN/VAE-based models? Do these extra samples work well with the virtual samples? Is the information redundant/complementary?

---

> ### Author Response · Authors · 2020-11-24
> **Responses to AnonReviewer2**
>
> 2.1 : “A large part of the improvement stems from the virtual classes, which is essentially mixup applied to ZSL. This is somewhat derivative.”
>
> We agree with the reviewer that mixup is a key component of our method. However, we point out that its effectiveness is largely enhanced by our proposed formulation. Our method is a two-expert (seen and unseen) approach, while the virtual classes/data by mixup are solely used in the episodic meta-learning for the unseen expert. On the other hand, simply applying mixup to other single-expert formulations most likely could not achieve comparable improvements in GZSL in that even with mixup, a single-expert classifier tends to bias the seen classes.
>
>
> 2.2: “How does the paper work with actual generated samples as in the GAN/VAE-based models? Do these extra samples work well with the virtual samples? Is the information redundant/complementary?”
>
> This is a reasonable suggestion. We need further experiments to assess if it could yield further improvements. We will include such discussions and additional experiments in the final version of our work.
>
> In the proposed episodic meta-learning for GZSL, we have the flexibility of working with not only virtual samples but also virtual classes. We find that for those datasets (e.g., AWA2) of a small number of classes, increasing the number of classes by mixup is advantageous for learning the S2V embedding. In comparison, the GAN/VAE approaches learn to generate samples for a fixed number of unseen classes.

---

### Official Review · AnonReviewer3 · 2020-10-30
**This paper describes a method for GZSL using a combination of adaptive and generative techniques to create virtual classes that help account for unseen classes.**

**Rating:** 7
**Confidence:** 5

**Review:**

Due to time shortage this will be a short review. I have gone through the paper carefully.
Motivation
The authors motivate their work well and have a thorough literature survey.
Method
The authors essentially create clusters that can anticipate unseen classes. That idea is not new in and of itself but the authors' realization of the idea through mix-ups and image-adaptive semantics is new and interesting. The overall approach is technically sound and each component is well motivated and described.

Results
The results are convincing. The authors get an across the board improvement over the state of the art.
Clarity
The paper is clearly written and has a good logical flow. I would recommend not using adjectives like elegant and insightful for one's own work. Perhaps such assessments are best left to the reviewers.
Quality, originality and significance
The paper presents moderate innovation in my view. It is thorough and hence does deserve consideration. In short, good quality, reasonable originality and decent significance.

---

> ### Author Response · Authors · 2020-11-24
> **Responses to AnonReviewer3**
>
> Thank you for the suggestion. We will improve the writing in the revision.

---

### Official Review · AnonReviewer5 · 2020-11-06
**Adaptive and Generative Zero-Shot Learning**

**Rating:** 6
**Confidence:** 5

**Review:**

The paper proposes a framework for the GZSL using the meta-learning and attention mechanism. The image-guided attention on the semantic space helps to adapt the better class specific semantic information. The modified semantic space projected to the visual space and in the visual space, cosine similarity is measured. The paper learns separate expert for the seen and unseen classes. The unseen class expert is trained with the pseudo negative samples with pseudo negative labels. Meta-learning based training helps to learn the model when only a few examples per class are available.

Comment:
1: The adaptive modification of the semantic space is novel in the ZSL setup with clear intuition. Also, the mixup idea in the ZSL setup seems to work well, it helps to separate the seen and unseen class data hence choosing the seen/unseen expert is easy.
2: The paper shows a strong result on the standard ZSL dataset. The ablation over the various component shows the efficacy of the proposed component.

3: How meta-learning is used in the paper is not clear also in the implementation details it's not mentioned. What is the task definition? How meta-train and meta-test are defined? The paper mention they use meta-learning, but nothing is clear about this. Therefore reproducibility is the main issue, and I think using the provided information the reported result can not be reproduced. Please update the experimental setup and provide the detail so that result can be easily reproduced.

4: There are a few recent works [a,b,c,d] that explore the meta-learning framework for the ZSL setup. The author should mention these paper and add the advantage of the proposed model over the other meta-learning approach also these result should be compared. Note: [d] is mentioned in the paper. Also please add an ablation if the model has advantage using the meta-learning, i.e. what is the result of the proposed component if we use the simple network without meta-learning.

[a] Meta-Learning for Generalized Zero-Shot Learning, AAAI-20
[d] Episode-based prototype generating network for zero-shot learning, CVPR-20
[b] Learning to Compare: Relation Network for Few-Shot Learning, CVPR-18
[c] Correction Networks: Meta-Learning for Zero-Shot Learning, ArXiv

5: Reproducibility is the main challenge here I request the author, please provide the code for the CUB dataset (since its look too high, I suspect the result in the inductive setting).

6: In the table-1, what is T for the ZSL result? Is it transductive setting?

Overall I like the idea, and paper shows a good result. Reproducibility is the main challenge, and CUB results look too high, I like to verify the result, therefore requesting the author to submit the code for the CUB dataset. Also, please discuss the suggested paper that is also using meta-learning based training. I will update my score on the successful verification of the code and result and the updated paper quality.

---

> ### Author Response · Authors · 2020-11-24
> **Responses to AnonReviewer5**
>
> 3. We have made our code available on github, https://github.com/anonmous529/AGZSL.
> In section 3.4, we describe how to apply episodic meta-learning to learn the unseen expert. Instead of forming each training episode based on samples from the seen classes, the meta-learning is carried out over virtual classes and their samples, which are obtained by the mixup scheme. (See equations (7) and (8).) More specifically, for each training episode, we use mixup to randomly generate 16  ~ 20 virtual classes and 4 samples for each class. In inference, the classification result by the seen and unseen experts is decided by equations (1) and (2).
>
>
> 4. Thank you for the information. In the revision, we have included them in the related work and the experiment comparisons, as listed in Table 1 of Section 4.
> Our method is a two-expert (seen and unseen) approach. Even the baseline model of our ablation study in Section 4.3 comprises two experts. If we exclude the unseen expert from the meta-learning, then the two-expert baseline in Table 2 will be reduced to a seen-expert baseline. As a result, take, for example, the baseline performance on CUB: The general unseen accuracy (U) will drop from 68.6 to 60.8 and the unseen accuracy (T) will drop from 73.7 to 69.4.
>
>
> 5. We have made our code available on github, https://github.com/anonmous529/AGZSL.
> In our experiments, we find that fine-tuning the backbone for feature representation would dramatically improve the GZSL performance on CUB.
>
>
> 6. In Section 4.1, the paragraph for Evaluation, we have stated that T denotes the average per-class top-1 accuracy. In the revision, we have changed it to "T1" to avoid possible confusions.

---

### Decision · Program_Chairs · 2021-01-07
**Final Decision**

**Decision:**

Accept (Poster)

**Comment:**

The submission combines meta-learning and attention mechanism for generalised zero-shot learning. The image-guided attention on the semantic space helps to adapt the better class specific semantic information while separate experts operate on the seen and unseen classes. The unseen class expert is trained with the pseudo negative samples with pseudo negative labels. Meta-learning based training adapts the model to few-shot learning scenario. The submission has received two accept, two weak accept and one weak reject reviews. All reviewers found the methodology interesting but they found it moderately novel. The experimental evaluation has been found strong. The rebuttal addressed all the reviewers' concerns and during the discussion phase all reviewers recommended acceptance. The meta reviewer follows the consensus of all the reviewers and recommends acceptance.